# Survey on Untethering of the Spinal Cord and Urological Manifestations among Spina Bifida Patients in Malaysia

**DOI:** 10.3390/children9071090

**Published:** 2022-07-21

**Authors:** Singh Nisheljeet, Abu Bakar Azizi, Kamalanathan Palaniandy, Dharmendra Ganesan, Teng Aik Ong, Azmi Alias, Ramalinggam Rajamanickam, Wahib M. Atroosh, Siti Waheeda Mohd-Zin, Andrea Lee-Shamsuddin, Singh Nivrenjeet, Warren Lo, Noraishah Mydin Abdul-Aziz

**Affiliations:** 1Invertebrate & Vertebrate Neurobiology Lab, Department of Parasitology, Faculty of Medicine, Universiti Malaya, Kuala Lumpur 50603, Malaysia; s2029043@siswa.um.edu.my (S.N.); mva190025@siswa.um.edu.my (S.W.M.-Z.); andyshamsuddinlee@gmail.com (A.L.-S.); p105905@siswa.ukm.edu.my (S.N.); 2Department of Surgery, University Kebangsaan Malaysia, Kuala Lumpur 56000, Malaysia; azizi@ppukm.ukm.edu.my (A.B.A.); pkpknathan@ppukm.ukm.edu.my (K.P.); 3Department of Surgery, Faculty of Medicine, Universiti Malaya, Kuala Lumpur 50603, Malaysia; dharmendra@um.edu.my (D.G.); ongta@ummc.edu.my (T.A.O.); 4Department of Neurosurgery, Tunku Abdul Rahman Neuroscience Institute (IKTAR), Hospital Kuala Lumpur, Kuala Lumpur 50586, Malaysia; azmidr@hotmail.com; 5Department of Pediatric Neurosurgery, Women and Children Hospital Kuala Lumpur, Kuala Lumpur 50586, Malaysia; 6Faculty of Law, Universiti Kebangsaan Malaysia, Kuala Lumpur 43600, Malaysia; rama@ukm.edu.my; 7Department of Parasitology, Faculty of Medicine, Universiti Malaya, Kuala Lumpur 50603, Malaysia; wahib@um.edu.my; 8Department of Urology, Kuala Lumpur Hospital, Jalan Pahang, Kuala Lumpur 50586, Malaysia

**Keywords:** spina bifida, urinary tract infection, antibiotic stewardship, urological management

## Abstract

The incidence and severity of urinary tract infections (UTIs) due to spina bifida is poorly understood in Malaysia. Tethering of the spinal cord is a pathological fixation of the cord in the vertebral column that can result in neurogenic bladder dysfunction and other neurological problems. It occurs in patients with spina bifida, and the authors of this study sought to investigate the impact of untethering on the urological manifestations of children with a tethered cord, thereby consolidating a previously known understanding that untethering improves bladder and bowel function. Demographic and clinical data were collected via an online questionnaire and convenient sampling techniques were used. A total of 49 individuals affected by spina bifida participated in this study. UTIs were reported based on patients’ observation of cloudy and smelly urine (67%) as well as urine validation (60%). UTI is defined as the combination of symptoms and factoring in urine culture results that eventually affects the UTI diagnosis in spina bifida individuals irrespective of CISC status. Furthermore, 18% of the respondents reported being prescribed antibiotics even though they had no history of UTI. Therefore, indiscriminate prescription of antibiotics by healthcare workers further compounds the severity of future UTIs. Employing CISC (73%) including stringent usage of sterile catheters (71%) did not prevent patients from getting UTI. Overall, 33% of our respondents reported manageable control of UTI (0–35 years of age). All individuals below the age of 5 (100%, *n* = 14) were seen to have improved urologically after the untethering surgery under the guidance of the Malaysia NTD support group. Improvement was scored and observed using KUB (Kidneys, Ureters and Bladder) ultrasound surveillance before untethering and continued thereafter. Spina bifida individuals may procure healthy bladder and bowel continence for the rest of their lives provided that neurosurgical and urological treatments were sought soon after birth and continues into adulthood.

## 1. Introduction

Spina bifida is a common congenital malformation that leads to lifelong complications involving the urinary tract [1]. Urological dysfunction is prevalent amongst patients with injuries pertaining to the spinal cord and is especially prevalent in children born with birth defects. In many countries, the major cause of death after one year of age has been attributed to urological dysfunction which leads to kidney failure [2,3]. There are various approaches to urological management in Malaysia for individuals with spina bifida from birth until the first symptoms arise due to urological morbidity and mortality. Urological difficulties are not only significant but prevalent for spina bifida individuals progressing into adulthood. Since it is important to improve the quality of life and maintain the general health of patients with spina bifida, an appropriate system of urological management is needed [4]. Despite this, Malaysia has yet to establish either a standard operating procedure for their urological management or a support system for general health of individuals with spina bifida from birth [5,6].

Furthermore, the use of prophylactic antibiotics is still debated in the care of individuals with spina bifida [7,8]. Although the formal health-related economic effects of Malaysians with spina bifida is not known, a simple estimation can be made for the minimal expenditure based on 8092 individuals. Taking into account an incidence of 1 in 1000 live births, the sample size was calculated if half of the Malaysian population born with spina bifida aperta (16,185 individuals) and half of those (8092) are individuals who survived as spina bifida aperta requiring surgery in Malaysian government/semi-government hospital setting, whereby the State bears the cost of RM 8000 per surgery. This in turn would cost the Malaysian government an estimated RM 64,740,000 for surgery on day one after birth alone (Ganesan, D., personal communication).

A study carried out on the survivability of patients with spina bifida has also shown that individuals affected by spina bifida develop end-stage renal disease (ESRD) at younger ages than patient without spina bifida. Urological issues were the most common primary cause of ESRD [9]. It is not known how many Malaysians suffer irreversible kidney damage due to under-supervised urological management brought upon by a birth defect. Individuals with spina bifida are at risk of progressive renal damage, in conjunction with recurrent urinary tract infections and a hostile neurogenic bladder. The impaired bladder may result in upper urinary tract deterioration, hydronephrosis, recurrent pyelonephritis and renal scarring [10]. Some individuals may progress to ESRD requiring dialysis. Hence, management of bladder function to prevent adverse urinary tracts infections to preserve renal function is critical.

Children with spina bifida are at high risk for urinary tract infections (UTI). However, there is no standardized definition of UTI, leading to variability in both clinical and research management. It is reported that the most commonly used definition included a combination of symptoms and culture results (34.5%), whereas 31% used a combination of symptoms, culture results, and urinalysis data [11,12]. In this study, UTI in spina bifida individuals has been defined as symptoms and urine culture results if available, irrespective of CISC (clean intermittent self-catheterization) status.

Antibiotic usage among Malaysians with spina bifida is not known and has never been investigated. The aim of this study was to investigate the experiences of individuals with spina bifida and their caregivers when faced with potential urinary tract infection and to explore the impact of untethering of the spinal cord on their urological management. Respondents were members of a Malaysian spina bifida support group known as Malaysia Neural Tube Defects (Malaysia NTD). However, this study does not analyze the result of the urine culture, nor does it measure in any way the patient response after appropriate antibiotic treatment.

## 2. Materials and Methods

### 2.1. Data Collection and Management

This study was carried out in accordance with the recommendation of the UMMC Medical Research Ethics Committee (MREC) with written informed consent from all respondents. Our study abided by the “Declaration of Helsinki” and informed consent was obtained from the respondents or their guardians before recruiting to the study and data collection. The study was reviewed and approved by the Medical Research and Ethics Committee (MREC (NMRR-14-1958-20549 (IIR)) of the Ministry of Health Malaysia. We advertised on social media platforms (WhatsApp and Facebook). A quantitative study aimed to spotlight the understanding of urological management among spina bifida patients within our cohort in Malaysia. A convenience sampling technique was defined; this included individuals who do ultrasound regularly, use CISC and validated their urine sample for UTI. A total of 49 individuals responded to the questionnaire (Appendix A).

### 2.2. Study Design

A survey was used to obtain the primary data to assess the level of understanding on urological management amongst spina bifida individuals in Malaysia. The patient population consisted of individuals with spina bifida of age group from 1 year old to 35 years old in Malaysia. The survey respondents were parents of spina bifida patients and spina bifida individuals.

Inclusion criteria are mothers who gave birth to infants who were born with spina bifida, mothers who were diagnosed with gestational diabetes, genetic disorders and family history of birth defects or diabetes. Exclusion criteria are spina bifida individuals with other co-morbidities (i.e., cardiovascular disease).

The guidance from Malaysia NTD support group helps to promote improved quality of life across the life span of individuals with spina bifida. Quality of life is defined as “an individual’s perception of their position in life in the context of the culture and value systems in which they live and in relation to their goals, expectations and concerns” [13,14]. The following practices have shown improvement among the patients in our cohort:I.Patient compliance in regards to neurosurgical and urological appointmentsII.KUB ultrasound has to be performed one month before and after the untethering surgery. Follow-up appointments are between three and six months depending on urological managementIII.In-depth explanation to the patient family in regards to the embryology of neural tube development followed by clear explanation on tethered cord syndrome and bladder dysfunction leading to kidney damage

### 2.3. Statistical Analysis

The data were analyzed statistically using statistical packages for social sciences (SPSS), version 26. A *p*-value of 0.05 was considered significant.

## 3. Results

In this study, 49 respondents were reported either as spina bifida individuals (6; 12%) or the parents (43; 88%) who answered the questionnaire on behalf of their spina bifida children. Amongst them, the most commonly reported NTD types were lipomyelomeningocele (Spina Bifida Occulta) (59%, *n* = 29) and myelomeningocele (Spina Bifida Aperta; 41% *n* = 20). The majority of spina bifida individuals reported in this study were non-syndromic (84%, *n* = 41), whereas only 16% (*n* = 8) were reported as syndromic spina bifida. There were in total 53% males (*n* = 26) and 47% females (*n* = 23) spina bifida patients with a mean of 9.76 years of age. In the 0–5 years age group, there was almost twice the number of males (*n* = 12) to females (*n* = 7). In the 6–20 years age group, there were more males (*n* = 11) compared to females (*n* = 8). Meanwhile, females were more (*n* = 6) than males (*n* = 1) in the 21–35 age group (Table A1).

A total of 28 (57%) individuals had undergone untethering surgery, of which 22 individuals (79%) were in the age group <5 years old and the remaining 6 respondents (21%) in the age group greater than 5 years of age (Table A2). The study revealed that 50% (*n* = 14) of the respondents who are in the age group <5 years old, underwent untethering surgery and subsequently had a significantly lower frequency of UTI compared to spina bifida individuals who are in the age group >5 years old. Meanwhile, only 7% (*n* = 2) in the age group >5 years said to have lowered their frequency of UTI after the untethering surgery (*p* = 0.009) (Table A4). On the other hand, 12 individuals (42.9%) reported no difference in the frequency of UTIs after the surgery. However, 18% (*n* = 8) of the respondents who had undergone untethering of the spinal cord surgery, reported zero occurrence of UTI which was validated by urine test and yet antibiotics were prescribed when they visited either the clinic or hospital. Furthermore, the study showed that 44.89% (*n* = 22) of all respondents claimed to have an improved kidney function after the surgery compared to 42.86% (*n* = 21) who did not undergo untethering surgery (*p* = 0.01) (Table A3).

Overall, this study has revealed that 53% (*n* = 24) of respondents claimed to have been prescribed antibiotics during their visit to the clinic (*p* = 0.033) including 4 individuals who did not have UTI but were prescribed antibiotics regardless and 71% (*n* = 32) of respondents claimed to have been prescribed antibiotics during their visit to the hospital (*p* = 0.003) including 6 individuals who did not suffer from UTI (Table A5). Besides, 67% of respondents who visited either the clinics or hospitals were experiencing UTI symptoms such as cloudy and smelly urine, in contrast to the 13 respondents who did not exhibit these symptoms (smelly/cloudy urine) (*p* = 0.057) (Table A5). In addition, 27% (*n* = 12) and 11% (*n* = 5) of the respondents claimed to have other symptoms of UTI such as stomach discomfort (*p* = 0.057) and back pain (*p* = 0.227) respectively.

Our study showed that 80% of the respondents had submitted their urine sample for culture and a significant number of patients who gave their urine samples at the hospitals (*p* = 0.003) or clinics (*p* = 0.033) were suffering from UTI before the untethering surgery (Table A5). However, this study does not analyze the result of the urine culture nor does it measure in any way the patient response after appropriate antibiotic treatment. This study also reported that 87% of respondents do kidney ultrasound regularly (*p* = 0.018) and the majority of patients were going directly to the clinic or the hospital during their first episode of UTI (*p* = 0.023). Around 73% of respondents practiced the use of clean intermittent self-catheterization (CISC) as a urological management amongst spina bifida in Malaysia, meanwhile, 24% of the respondents did not employ CISC (*p* = 0.514). Considering attitudes towards CISC, 69% of respondents said CISC is an invasive procedure, while 27% of the respondents were not sure about the invasiveness of the CISC procedure. The degree of sterility practiced was also a major concern; 71% of the respondents used sterile catheters during every CISC procedure, whereas 29% of respondents did not employ any means of sterile catheterization.

## 4. Discussion

In Malaysia, clinics are ill-equipped to deal with complications arising from spina bifida. This is supported by the finding that the clinic-goers among spina bifida individuals in our cohort were not attending clinics specifically for the complaint of UTI. Interestingly, a significant value was obtained (*p* = 0.033) irrespective of health complaint for handing urine samples to clinics (Table A5). The justification from this statement comes from the fact that clinics prescribe antibiotics without knowing which antibiotic to prescribe to the spina bifida patient irrespective of whether the above-said urine was cultured or not. Our data showed that 18% of the respondents who were given their urine for validation of UTI, were in fact ill-advisedly prescribed antibiotics for UTI, even though they had no history of suffering from urinary tract infections. Urine validation here is defined as the combination of symptoms and culture results that affects the UTI diagnosis.

According to the Ministry of Health Malaysia, spina bifida is recognized primarily as a neurological disorder requiring the care of a neurosurgeon [1] but urologists equipped to deal with a functional bladder in Malaysia are scarce. Micturition requires proper function of the bladder and urethral sphincter complex. This process requires coordination between the central and the peripheral nervous system. Disruption of this pathway caused by damage or disease can lead to the neurogenic bladder (NGB) with abnormal bladder storage and emptying [2]. Our data showed that UTI is significantly prevalent before the untethering surgery (Table A5) but improved after untethering whereby frequency of UTI was lowered (*p* = 0.009) (Table A4). The urologic representation of a patient’s left kidney as seen in Figure A1, shows pre and post impact of untethering of spinal cord on kidney condition. Susceptibility to UTI appears when there is retention of urine in the bladder due to lack of bladder function and stimulus leading to overactivity of detrusor muscle and vesicoureteral reflux. Backflow of urine damages the kidney and causes a build-up of bacteria due to prolonged urine retention which eventually causes infection and fever [8]. Untethering of the spinal cord as seen in Figure A2, helps to release the pressure of the tethered nerves, and stimulates the bladder for better voiding and functional recovery [14,15]. The indication for the decision to undertake the tethered cord release is urine retention in the bladder, which can be observed under a KUB ultrasound. UTI in patients with neurogenic bladders remains a challenge for urologists and other clinicians. This is due to the fact that altered anatomy and bladder mechanics, frequent catheterization, impaired sensation, and immunologic dysfunction make patients with neurogenic bladder uniquely prone to UTIs, which are difficult to diagnose and treat appropriately. Although advances in medical care have significantly reduced the morbidity and mortality of urinary tract infection (UTI) in patients with neurogenic bladder (NGB), ill-advised use of antibiotics showed 10% to 15% of patients with NGB die from sepsis of urinary origin [15,16]. However, having the correct urological surgical intervention can prevent infections and lead to a better quality of life as well as ensuring a lifetime with a healthy pair of kidneys.

According to our study, Malaysians probably do not know how to carry out CISC properly, although the use of sterile catheters shows a better prognosis than re-using catheters. There should not be any problem for spina bifida patients to have healthy, functioning kidneys their whole life, but deterioration of kidneys is often seen in spina bifida patients (most likely since the majority of Malaysian spina bifida patients do not see a urologist). Most spina bifida patients in Malaysia are referred to nephrologists instead. However, the reasons for this remains unanswered. The question most asked by parents of babies with spina bifida to their urology consultant is whether they need to employ CISC in the management of their child’s urological needs. The second question is when do they need to begin and the third question asked is how often does CISC need to be carried out [17]. In most clinical settings, the question of whether there could be other means apart from CISC more well suited for the spina bifida child is often not asked nor investigated. A study carried out on family compliance towards CISC shows that although most parents dislike CISC, they complied with the suggested management [16,17].

Furthermore, a study [17] on the association between UTI and CISC showed that infants and toddlers with spina bifida who were initially managed with spontaneous voiding had a lower risk of UTI than those managed immediately with CISC (irrespective of bladder condition) and this is consistent with our study which found that CISC usage did not significantly reduce the recurrence of UTI (*p* = 0.514). However, patients who do CISC after a period of voiding did not have a significantly different risk of UTI which suggest that early initiation of CISC may not be warranted in all infants with spina bifida [17,18]. In fact, CISC at an early age may potentially have a detrimental effect on the child such as trauma, loss of sensory control and reflexes [18]. CISC [19], which has been a foundation used by most urologists and other clinicians as a modality for urological management amongst spina bifida individuals with neurogenic or atonic bladders encompassing urinary retention, incontinence and infection needs to be re-assessed. The Lapides technique [20] need not necessarily be the gold standard with the advent of science and technology. In recent studies [16,21,22], infants and toddlers with spina bifida initially treated with spontaneous urination had a lower risk of UTI than those treated with CISC. Patients who switched to CISC after the initial observation period with urination did not have a significantly different risk of UTI than those treated with CISC alone. These results suggest that a reduction in UTI risk through early CISC use may not be guaranteed in all infants with spina bifida. Quality of life of spina bifida children was markedly improved by untethering of the spinal cord, as our findings suggest that 100% (*n* = 14) of respondents in the age group zero to five years old had lowered their UTI frequency after the untethering of the spinal cord (Table A3) especially if performed at an early age [23,24]. This is further strengthened as our data showed that 45% of the respondents had improved their kidney function after the untethering surgery (*p* = 0.01). Moreover, an important consideration is that families with budget constraints may not be able to buy catheters or afford advice from clinics of hospitals, especially outside of Kuala Lumpur [25,26].

Respondents who took antibiotics during their visit to the clinic or hospital were in fact suffering urinary tract infection (UTI) (Table A5), whereby 82% of the respondent’s attended clinic/hospital when they first suspected UTI. This suggests that individuals with UTIs in Malaysia are able to manage and mobilize themselves to get the correct treatment. On average there was an increase of 500 g of urine in the span of 4–6 h upon untethering of spinal cord and frequency of diapers changed increased as well (personal communication with the respondents). Before the untethering surgery, families typically changed diapers every 4 h, which equates to a total of six diaper changes per day. In comparison to after the untethering surgery, on average, the diapers were changed every 2 h which meant doubling the total number of diaper change. This suggests that untethering of the spinal cord promotes a higher volume of urine voiding. The main goal of urological treatment in children with spina bifida is to reduce the risk of urinary tract infections (UTIs) and associated renal disorders. CISC has been the mainstream of treatment, but recent studies have shown that this approach is not without risk [27]. However, this study is limited by a larger dataset that would provide a more convincing prevalence association between untethering the spinal cord at an early age and a better prognosis on the urological manifestations after untethering.

## 5. Conclusions

Although asymptomatic bacteriuria should not be treated in most patients, the lack of a standardized management of UTI has greatly hindered attempts to optimally diagnose, treat and investigate recurrent UTIs in patients with neuropathic bladder. This area requires a great deal of understanding and continuous efforts. Based on this pilot study, Malaysia’s guidelines for consistent antibiotic management appear not to exist, as respondents reported that antibiotics were prescribed without probable cause. Therefore, it is crucial to set the standard of care for patients with spina bifida and pursue a larger data set that provides a more compelling prevalence association between spinal cord suppression at an early age and a better prognosis for urinary treatment.

It is alarming that this study found that patients who went to the clinic did not receive the proper care needed to maintain urological health. Hospitals that process urine for culture purposes are advised to establish standard treatment for patients with spina bifida. Education for both patients and medical staff is paramount to carrying this out effectively. More importantly, our results support that untethering of the spinal cord in individuals with spina bifida before the age of two years improves urological function, and this finding should be broadly pursued with appropriate government funding on a larger scale.

## Data Availability

The data presented in this study are available on request from the corresponding author. The data are not publicly available due to concerns regarding participant/patient anonymity.

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
