# Peer review of "Survey on Untethering of the Spinal Cord and Urological Manifestations among Spina Bifida Patients in Malaysia"

_children, 2022, doi:10.3390/children9071090_

Round 1

Reviewer 1 Report

I think the authors may want to change the paper to be more of a survey results of urologic management of patients with spina bifida in Malaysia. This would include antibiotic usage and CIC frequency patient reported symptoms etc. It's hard to draw some of the conclusions they are making as the definition of UTI (smelly cloudy urine) is not necessarily correct. There is no info on fever flank pain hematuria etc. and culture results are limited. How are they proving that kidney function is improved after tethered cord release?  If the definition of UTI isn't really clear then its hard to make the conclusion that tethered cord release leads to less UTI. They report 57% of their subjects had tethered cord release. That is a very high number what were the indications for those releases. Also looking at the figure of the MRI they provided while some of the lipoma has been removed the cord still seems tethered to me on the image.

Thus I think I might suggest to the authors that they use this survey data to report on the current state of urologic management of SB patients in this support group in Malaysia including antibiotic use

Author Response

Response to Reviewer 1 and 2

20th June 2022

Dear reviewers,

            Thank you for the review comments of the manuscript. We have addressed the comments from Reviewer 1 & 2. All the amendments made in the revised manuscript with track changes are shown in red and bold fonts. The line numbers are based on ‘No Markup’ in Display for Reviews’ in the ‘Track Changes’ tool bar function on the revised manuscript with track changes. The list of references were also revised accordingly.

Kind regards,

Dr Noraishah Mydin Abdul-Aziz

Universiti Malaya, Kuala Lumpur, Malaysia

Response to reviewers

Reviewer 1

Point 1: I think the authors may want to change the paper to be more of a survey result of urologic management of patients with spina bifida in Malaysia. This would include antibiotic usage and CIC frequency patient reported symptoms etc. It's hard to draw some of the conclusions they are making as the definition of UTI (smelly cloudy urine) is not necessarily correct. There is no info on fever flank pain hematuria etc. and culture results are limited. How are they proving that kidney function is improved after tethered cord release?  If the definition of UTI isn't really clear then it’s hard to make the conclusion that tethered cord release leads to less UTI.

Response 1: We thank Reviewer 1 for the thorough recommendation and the sentence has been revised to reflect our findings. Urinary tract infections in our pretext are diagnosed based on patients reporting cloudy and smelly urine (67%) and urine validation (60%). Since we do not have other data defining the causality of UTI from spina bifida patients, therefore cloudy and smelly urine was utilized as one of our variables. Also, the occurrence of UTI amongst spina bifida patients coincides with either single or multiple use of catheter (Farrelly E. et al., 2020) (Line 86-94).

“Children with spina bifida are at high risk for urinary tract infections (UTI). However, there is no standardized definition of UTI, leading to variability in both clinical and research management respectively (Foster et al., 2021). In the same study, it is reported that the most commonly used definition included a combination of symptoms and culture results (34.5%), whereas 31% used a combination of symptoms, culture results, and urinalysis data. Hence, our data supports the symptoms of UTI which are cloudy and smelly urine. Besides, urinary tract infections are more common in children with spina bifida (SB) than in neurologically intact children, and E. coli is the most common urinary pathogen in the general paediatric population (Ortiz et al., 2018)”

Reference

  1. Farrelly E, Lindbo L, Wijkström H, Seiger Å. The Stockholm Spinal Cord Uro Study: 2. Urinary tract infections in a regional prevalence group: frequency, symptoms and treatment strategies. Scand J Urol. 2020 Apr;54(2):155-161. doi: 10.1080/21681805.2020.1734078. Epub 2020 Mar 9. PMID: 32148149.
  2. Forster CS, Kowalewski NN, Atienza M, Reines K, Ross S. Defining Urinary Tract Infections in Children With Spina Bifida: A Systematic Review. Hosp Pediatr. 2021 Nov;11(11):1280-1287. doi: 10.1542/hpeds.2021-005934. PMID: 34697071.
  3. Ortiz TK, Velazquez N, Ding L, Routh JC, Wiener JS, Seed PC, Ross SS. Predominant bacteria and patterns of antibiotic susceptibility in urinary tract infection in children with spina bifida. J Pediatr Urol. 2018 Oct;14(5):444.e1-444.e8. doi: 10.1016/j.jpurol.2018.03.017. Epub 2018 Apr 20. PMID: 29709445.

Point 2: They report 57% of their subjects had tethered cord release. That is a very high number what were the indications for those releases. Also looking at the figure of the MRI they provided while some of the lipoma has been removed the cord still seems tethered to me on the image. Thus, I think I might suggest to the authors that they use this survey data to report on the current state of urologic management of SB patients in this support group in Malaysia including antibiotic use

Figure C2. MRI scan of lumbar spine sagittal view showing findings of tethered spinal cord. (A) Tethered cord at Lumbar 1 and Lumbar 2 (L1/L2) before untethering surgery (B) Post untethering surgery shows release of tethered cord.

Response 2: We thank reviewer 2 for the thorough check. Tethered cord release is the release of tension upon the cord according to our neurosurgeons. Probably what reviewer 2 is saying is about the physical release which is almost always associated with morbidity. Amongst the individuals with spina bifida in our cohort, we focused on reducing the tension of the spinal cord for functional recovery. In fact, a recent case study (Harazeen et al., 2022) of tethered cord release with complete lipoma excision showed that the patient’s symptoms did not improve. In the Harazeen et al (2022) paper, the urologist was also consulted and recommended self-catheterization using clean intermittent self-catheterization (CISC) every six hours, whereas in our cohort of untethered individuals, we see improvement in bladder and bowel function. So, complete excision of lipoma MAY NOT improve patient’s condition, functionally.

Reference:

  1. Harazeen A, Thottempudi N, Sonstein J, Li X, Wu L, Rai P, Masel T. Tethered Cord Syndrome Associated With Lumbar Lipomyelomeningocele: A Case Report. Cureus. 2022 Feb 25;14(2):e22590. doi: 10.7759/cureus.22590. PMID: 35355549; PMCID: PMC8957751.

Reviewer 2 Report

The Introduction section is well written with adequate literature.

The aim of the study is not well defined. Please reformulate the aim.

In methods section explain how did you classified the age groups. Additionally, it is not clinically relevant to perform the study on entire age group. Small children and their caregivers respond different than adults. What type of spina bifida was investigated and it is not clear whether the spina bifida was the cause of the infection. Please insert exclusion criteria such as joined anomalies including reflux etc....

In results section please add in first table joined anomalies if any, type of spina bifida.

In discussion section the limitations part lacks.

The conclusions should be rewritten and adjusted to the aim bearing in mind the number of patients. For example it is not quite possible to draw firm conclusion such as the last sentence in the Conclusion part.

Author Response

Response to Reviewer 2

20th June 2022

Dear reviewers,

            Thank you for the review comments of the manuscript. We have addressed the comments from Reviewer 1 & 2. All the amendments made in the revised manuscript with track changes are shown in red and bold fonts. The line numbers are based on ‘No Markup’ in Display for Reviews’ in the ‘Track Changes’ tool bar function on the revised manuscript with track changes. The list of references were also revised accordingly.

Kind regards,

Dr Noraishah Mydin Abdul-Aziz

Universiti Malaya, Kuala Lumpur, Malaysia

Reviewer 2

Point 1: The Introduction section is well written with adequate literature.

Response 1: We thank Reviewer 2 for the thorough check.

Point 2: The aim of the study is not well defined. Please reformulate the aim.

Response 2: The comment has been addressed as below

To investigate the impact of untethering on the urological outcome of children with a tethered chord, thereby identifying proper urological care for individuals with spina bifida.”

Point 3: In methods section explain how did you classified the age groups. Additionally, it is not clinically relevant to perform the study on entire age group. Small children and their caregivers respond different than adults. What type of spina bifida was investigated and it is not clear whether the spina bifida was the cause of the infection. Please insert exclusion criteria such as joined anomalies including reflux etc....

Response 3: The comment has been addressed as below

“A survey employed to obtain the primary data to assess the level of understanding and confirmation on urological management amongst spina bifida individuals in Ma-laysia. Source of population were individuals with spina bifida of age group from 1 years old to 35 years old in Malaysia. Study population are women who are re-productively active, parents of spina bifida patients and spina bifida individuals.

Inclusion criteria are mothers who gave birth to infants who were born with spina bifida, mothers who were diagnosed with gestational diabetes, genetic disorders and family history of birth defects or diabetes. Exclusion criteria are spina bifida individuals with other co-morbidities (i.e., cardiovascular disease)” (Line 126-134).

Point 4: In results section please add in first table joined anomalies if any, type of spina bifida.

Response 4: The comment has been addressed as below

Table B1. General characteristics of respondents (n=49).

Description

n (%)

Participants:

43 (88%)

Parents of spina bifida Individuals

43 (88%)

Spina Bifida Individuals

6 (12%)

Type of Spina Bifida

Occulta

29 (59%)

Aperta

20 (41%)

Anomalies

Syndromic

8 (16%)

Non-syndromic

41 (84%)

Gender:

Male

26 (53%)

Female

23 (47%)

Age group:

Male

Female

0-5 years

12 (24%)

7 (14%)

6-20 years

13 (27%)

9 (16%)

21-35 years

1 (2%)

7 (14%)

            “Amongst them, the most commonly reported NTD type was lipomyelomeningocele (Spina Bifida Occulta) (59%, n=29) and 41% (n=20) reported with  myelomeningocele (Spina Bifida Aperta). Besides, majority of spina bifida individuals reported in this study were non-syndromic (84%, n=41), whereas only 16% (n=8) were reported as syndromic spina bifida.” (Line 141-145)

Point 5: In discussion section the limitations part lacks.

Response 5: The comment has been addressed as below

“In a recent study, infants and toddlers with spina bifida initially treated with spontaneous urination had a lower risk of UTI than those treated with CISC. Patients who switched to CISC after the initial observation period with urination did not have a significantly different risk of UTI than those treated with CISC alone. These results suggest that early start of CISC may not be guaranteed in all infants with spina bifida (Line 250)

“The main goal of urological treatment in children with spina bifida is to reduce the risk of urinary tract infections (UTIs) and associated renal disorders. Clean intermittent catheterization (CISC) has been the mainstream of treatment, but recent studies have shown that this approach is not without risk. However, this study is limited by a larger dataset that would provide a more convincing prevalence association between untethering the spinal cord at an early age and a better prognosis on the urological outcome.” (Line 273-279).

Reference

  1. Kaye IY, Payan M, Vemulakonda VM. Association between clean intermittent catheterization and urinary tract infection in infants and toddlers with spina bifida. J Pediatr Urol. 2016 Oct;12(5):284.e1-284.e6. doi: 10.1016/j.jpurol.2016.02.010. Epub 2016 Mar 4. PMID: 27118581

Point 6: The conclusions should be rewritten and adjusted to the aim bearing in mind the number of patients. For example it is not quite possible to draw firm conclusion such as the last sentence in the Conclusion part.

Response 6: The comment has been addressed as below

“Although asymptomatic bacteriuria should not be treated in most patients, the lack of a standardized management of UTI is greatly hindered to optimally diagnose, treat and investigate recurrent UTI in patients with neuropathic bladder. This area requires a great deal of understanding and continuous efforts. Based on this pilot study, Malaysia's antibiotic management is believed to be almost completely non-existent, as none of the respondents reported that the problem of urine culture analysis had to be addressed. Therefore, UTI prophylaxis in this scenario is considered important. It is crucial to set the standard of care for patients with spina bifida and pursue a larger data set that provides a more compelling prevalence association between spinal cord suppression at an early age and a better prognosis for urinary treatment.

Besides, it is alarming that this study found that patients who went to the clinic did not receive the proper care needed to maintain urological health. Hospitals that process urine for culture purposes are advised to establish standard treatment for patients with spina bifida. Education for both patients and medical staff is paramount to doing this effectively. More importantly, our results support that untethering of the spinal cord in a spina bifida individual before the age of 2 improves the urological function, and this finding should be broadly pursued with appropriate government funding on a larger scale.” (Line 284-301)

Round 2

Reviewer 2 Report

The manuscript is suitable for acceptance.

Author Response

Response to Reviewer 2

10th July 2022

Dear reviewers,

            Thank you for the review comments of the manuscript. We have addressed the comments from Reviewer 1 & academic editor. All the amendments made in the revised manuscript with track changes are shown in red and bold fonts. The line numbers are based on ‘No Markup’ in Display for Reviews’ in the ‘Track Changes’ tool bar function on the revised manuscript with track changes. The list of references was also revised accordingly.

Kind regards,

Dr Noraishah Mydin Abdul-Aziz

Universiti Malaya, Kuala Lumpur, Malaysia

Response to reviewer 2

Point 1: The manuscript is suitable for acceptance

Response 1: We thank Reviewer 2 for the thorough suggestions and accepting our manuscript.
